# MASLD and the Development of HCC: Pathogenesis and Therapeutic Challenges

**DOI:** 10.3390/cancers16020259

**Published:** 2024-01-06

**Authors:** Anju G. S. Phoolchund, Salim I. Khakoo

**Affiliations:** Faculty of Medicine, University of Southampton, Southampton General Hospital, Tremona Road, Southampton SO16 6YD, UK

**Keywords:** non-alcoholic fatty liver disease (NAFLD), metabolic-dysfunction-associated steatotic liver disease (MASLD), cancer, liver, hepatocellular carcinoma (HCC), immunotherapy

## Abstract

**Simple Summary:**

Metabolic-dysfunction-associated steatotic liver disease (previously known as non-alcoholic fatty liver disease) is a term for a range of liver conditions in which excess fat builds up in the liver. This can eventually lead to liver inflammation, chronic liver disease and, in some cases, a form of liver cancer (hepatocellular carcinoma). This condition is increasing worldwide, with patients most at risk being overweight or obese or having type 2 diabetes. Hepatocellular carcinoma is seen in patients with this condition at different stages, and research is ongoing to identify why some patients appear to develop cancer at an earlier stage of liver disease than others, and how to best treat them, as there appear to be variations in how well patients respond depending on what caused the liver inflammation initially. Here, we summarise the research to date and discuss the potential rationale for what has been observed.

**Abstract:**

Metabolic-dysfunction-associated steatotic liver disease (MASLD, previously known as non-alcoholic fatty liver disease (NAFLD)) represents a rapidly increasing cause of chronic liver disease and hepatocellular carcinoma (HCC), mirroring increasing rates of obesity and metabolic syndrome in the Western world. MASLD-HCC can develop at an earlier stage of fibrosis compared to other causes of chronic liver disease, presenting challenges in how to risk-stratify patients to set up effective screening programmes. Therapeutic decision making for MASLD-HCC is also complicated by medical comorbidities and disease presentation at a later stage. The response to treatment, particularly immune checkpoint inhibitors, may vary by the aetiology of the disease, and, in the future, patient stratification will be key to optimizing the therapeutic pathways.

## 1. Introduction

Non-alcoholic fatty liver disease (NAFLD) is an encompassing term for a spectrum of chronic liver diseases, ranging from simple steatosis of at least 5% in an imaging or histological assessment of the liver (also known as non-alcoholic fatty liver (NAFL)) to non-alcoholic steatohepatitis (NASH), which is characterised by inflammation associated with steatosis, to the development of fibrosis and established cirrhosis. A key complication of NAFLD is the development of hepatocellular carcinoma. The term metabolic-dysfunction-associated steatotic liver disease (MASLD) has now replaced NAFLD [1,2]. MASLD is defined as the presence of hepatic steatosis in conjunction with at least one cardiometabolic risk factor, while the presence of steatohepatitis is now termed metabolic dysfunction-associated steatohepatitis (MASH). This review will use the newly accepted MASLD/MASH nomenclatures.

## 2. Epidemiology of MASLD

Obesity rates are rising, having almost tripled since 1975, with 39% of adults estimated by the WHO to be either obese or overweight in 2016 [3]. MASLD rates mirror the increasing incidence of obesity, and represents a major cause of chronic liver disease worldwide, with a recent meta-analysis reporting an overall global prevalence of 30%, increasing by 50% between 1990 and 2019 [4]. The rates by country have been reported as 33% in SE Asia and South Asia, 44% in Latin America and 25% in Western Europe [4]. The true prevalence is likely underestimated in the literature due to the previous practice of diagnosing “cryptogenic cirrhosis” in the absence of evidence of viral, alcohol or immune-related liver disease. This increase is also further demonstrated in a recent European liver transplant registry analysis showing an increase in the percentage of recipients transplanted for MASH-related complications from 1.2% in 2002 to 8.4% in 2016 [5].

## 3. Pathogenesis and Progression of MASLD

MASLD is characterised by excessive lipid accumulation associated with insulin resistance in patients for whom alternative causes, such as viral hepatitis, significant alcohol intake and other secondary causes, have been excluded. There is a strong and complex relationship between MASLD and metabolic syndrome, which is defined as the combination of insulin resistance/type 2 diabetes mellitus, obesity, hypertension and dyslipidaemia [6]. It is being recognised as a complex multi-system disorder, and MASLD is increasingly being described as its hepatic manifestation. One study of patients with MASH observed that, within this population, the prevalence of obesity was 82%, that of type 2 diabetes mellitus was 47%, that of hyperlipidaemia was 72% and that of metabolic syndrome was 71% [7]. Type 2 diabetes mellitus in MASLD has been recognised as a risk factor for progression to MASH, cirrhosis and mortality, and poor glycaemic control is associated with a higher likelihood of MASH and/or advanced fibrosis [8,9].

The pathological processes behind MASLD have been described through the concept of a “multiple hit” hypothesis that asserts that genetically susceptible patients are impacted by environmental factors (lifestyle, gut microbiome, dietary choices, obesity) leading to lipid accumulation, insulin resistance, obesity and unfavourable alterations in the gut microbiota [10]. This, in turn, leads to increased hepatic de novo lipogenesis and the impaired inhibition of adipose tissue lipolysis, resulting in the increased delivery of free fatty acids to the liver and the accumulation of hepatic fat. The subsequent lipotoxicity contributes to mitochondrial dysfunction, with the production of reactive oxygen species and endoplasmic-reticulum stress. This is compounded by insulin resistance and increased levels of the absorption of lipopolysaccharides (LPSs) from the gut due to increased gut permeability secondary to the abovementioned altered gut microbiome. Cellular damage can then trigger immune cell infiltration, fibrogenesis and subsequent hepatic progenitor cell activation. Insulin resistance additionally results in adipose tissue dysfunction via the secretion of adipokines and proinflammatory cytokines, thereby potentiating inflammation [9]. At the cellular level, these processes result in oxidative stress and DNA damage in hepatocytes, which, combined with inflammation, is thought to eventually lead to fibrosis and cirrhosis. This has been reviewed in more detail elsewhere [11,12].

The natural history of MASLD is still not fully understood and is likely influenced by multiple extra-hepatic factors, as described above. Chronic hepatic inflammation is a key step leading to the development of MASH from “simple steatosis” and can be important in the development of HCC [13]. Only a minority of patients with steatosis will develop MASH, which is thought to be the key determinant in the further progression towards bridging fibrosis and subsequent liver-related complications/liver-related deaths [14]. However, it is worth noting that, in keeping with the concept of the multi-system effects of metabolic syndrome, cardiovascular-disease-related death has been shown to be the main cause of death in MASLD [15]. The progression from simple steatosis to more severe forms of MASLD does not appear to be a linear, unidirectional relationship. Paired-biopsy studies have demonstrated a dynamic relationship between MASLD and MASH over time, with a large single-centre study of 108 patients over a median of 6.7 years reporting that 42% of them showed signs of progression, 40% demonstrated stable disease and 18% demonstrated regression from either condition [16].

The management of MASLD includes lifestyle modifications to reduce weight, including dietary modifications and exercise [17,18,19]. These have limited benefits in many individuals and therefore therapeutic strategies are also important. Diabetes mellitus should be controlled and there is currently a great deal of interest in GLP-1 agonists, such as liraglutide and semaglutide [20,21,22,23,24,25].

## 4. Hepatocellular Carcinoma in MASLD

Hepatocellular carcinoma (HCC) accounts for 90% of primary liver cancer. According to 2020 estimates, primary liver cancer is the sixth most common cancer worldwide and the third most common cause of cancer-related death [26]. HCC most commonly develops against the backdrop of chronic liver disease with varying aetiologies. While, previously, chronic viral hepatitis was the most common aetiology of chronic liver disease leading to HCC [27], due to the high prevalence of chronic HBV in sub-Saharan Africa and Southeast Asia, successful treatment and prevention programmes for viral hepatitis have reduced the prevalence of these in recent decades [28,29]. Simultaneously, the rates of MASLD have increased significantly, as described above. A large U.S. healthcare database study of 4406 reported HCC cases identified MASLD as the most common form of chronic liver disease at 59% [30]. Mortality from HCC remains high, with 5–15% survival at 5 years [31]. As the prevalence of MASLD continues to rise, so do the rates of MASLD-HCC, with one study demonstrating an increase of 9% per annum between 2004 and 2009 in the U.S. [32]. MASLD-HCC is now the fastest rising cause of HCC and the fastest growing indication for orthoptic liver transplantation (from 2.1% to 16.2% from 2000 to 2016) [33]. Dynamic Markov modelling for MASLD-HCC across eight countries predicts a 122% rise in its incidence by 2030 [34]. The incidence of HCC in patients with established MASLD cirrhosis ranges from 0.7 to 2.6%.

There is a significant overlap between the recognised risk factors for the progression of MASLD to advanced fibrosis and those that are associated with subsequent tumorigenesis. Key risk factors for MASLD-HCC development are age, male gender and the presence of advanced fibrosis/cirrhosis [32,35]. As described earlier, metabolic syndrome is strongly associated with the development of MASLD. The components of this clinical phenotype have, in turn, been shown to be independent risk factors for HCC, particularly obesity and diabetes mellitus [36]. Obesity is an independent risk factor for multiple forms of malignancy [37], including HCC [38]. Additionally, genetic polymorphisms, especially those related to PNPLA3, may be a contributory factor to disease progression in MASLD as well as, in some cases, MASLD-HCC, as illustrated in Table 1.

The presence of cirrhosis remains the most important risk factor for the development of HCC, with a more than 10-fold increase in the risk of HCC with progression to cirrhosis [39]. However, there is increased recognition that MASLD-HCC can develop at an earlier stage, with up to 25–50% of cases of MASLD-HCC arising in patients without established cirrhosis, as illustrated in Table 2, which summarises recent studies on the rates of HCC within cohorts of patients with MASLD.

**Table 1 cancers-16-00259-t001:** Genetic polymorphisms associated with MASLD and MASLD-HCC.

Gene	Activity	Relevant Polymorphism	Function	Frequency	Relationship with MASLD Progression	Relationship with Development of HCC
*PNPLA3* (Patatin-like phospholipase domain containing 3)	Hydrolyses triglycerides and retinyl esters	rs738409 c.444 C > G, p.I148M	Encodes a methionine substitution that delays proteasomal degradation and hampers lipid mobilisation	17–49% [40]	Increased risk of MASLD, MASH and fibrosis [41,42]	Independent risk factor for HCC [43,44,45], independent of gender, age, BMI, T2DM and presence of advanced fibrosis/cirrhosis
*MBOAT7* (Membrane bound 0-acetyl transferase domain containing 7)	Phospholipid remodelling gene	rs641738 C > T	Reduces expression of hepatic MBOAT7 protein, favouring fat accumulation	35–40%	Increased hepatic fat content, MASLD, MASH and fibrosis [46,47]	Associated with HCC, independent of presence of cirrhosis [48]
*TM6SF2* (Transmembrane 6 superfamily member 2)	Role in triacylglycerol-rich lipoprotein lipidation	rs58542926 c.449 C > T, p.Glu167Lys	Retention of VLDL in the liver	3.4–7.2%	Increased hepatic TG content, MASH and fibrosis [49,50]	Not significantly associated with HCC in multivariate analysis [49]

Abbreviations: HCC, hepatocellular carcinoma; MASLD, metabolic dysfunction associated steatotic liver disease; T2DM, type 2 diabetes mellitus.

Looking more closely at the non-cirrhotic population, the risk of HCC has been shown to be higher in individuals with inflammation than in those without it, at 5.29 per 1000 p.a. in MASH vs. 0.44 per 1000 p.a. in MASLD [7]. However, there are relatively few studies that have determined the true incidence of HCC in pre-cirrhotic MASH due to the need for a liver biopsy to confirm the stage of the disease. Table 3 compares subpopulations of patients with and without MASLD cirrhosis in recently published cohorts of patients with HCC.

## 5. Pathogenesis of MASLD-HCC

The pathogenesis of HCC in MASLD is complex and still being elucidated. An outline of the potential pathways is illustrated in Figure 1. The lipid accumulation in hepatocytes and associated lipotoxicity create a dynamic proinflammatory environment and eventually lead to fibrogenesis [59]. The ongoing hepatocyte regeneration and tissue remodelling lead to an increased risk of subsequent tumorigenesis. Multiple oncogenic pathways within this proinflammatory environment are implicated in the development of HCC, impacting the genomic stability and telomere maintenance, as well as leading to alterations in DNA damage response pathways and aberrant signal transduction cascades [60,61,62]. The wide variety of mechanisms of tumorigenesis results in significant heterogeneity in HCC.

### 5.1. Immune Response in MASLD and HCC

The liver is uniquely immunotolerant, which is important to limit inflammation secondary to ongoing antigen exposure via the portal circulation. The mechanisms that maintain the homeostasis and immune tolerance are disrupted in the setting of MASLD and chronic inflammation. Both the adaptive and innate arms of the immune system are involved.

In a mouse model of MASH, activated CD8+ T cells and NKT cells accelerated the disease progression and hepatic tumorigenesis through the secretion of proinflammatory molecules [63]. A second murine study demonstrated that increased CD8+ PD1+ T cells impairs immunosurveillance and triggers hepatocarcinogenesis [64]. However, it appears that a subset of CD8+ T cells also plays a protective role, supporting the resolution of inflammation in the resolution of MASH in mice [65]. Another dysregulated T-cell group is CD4 T cells, which usually support efficient immune surveillance and impair tumorigenesis but are reduced in MASH [66].

Dysfunction of the innate immune system is another key player in MASH development. In the setting of MASH, natural killer (NK) cells appear to be more activated, although there are contradictory studies regarding the changes in the frequencies of the circulating cell numbers [67,68]. Kupffer cells, which function as resident macrophages of the liver, are activated in the setting of hepatic inflammation, increasing the expressions of a range of proinflammatory cytokines, which contribute to further hepatocyte inflammation and hepatic stellate cell activation. Thus, an overall ineffective immune response appears to promote chronic inflammation and hepatocarcinogenesis, whilst an effective immune response can potentially clear malignant hepatocytes [69].

### 5.2. Signalling Pathway Deregulation

Several signalling pathways have been reported to be deregulated in HCC. These are the subject of considerable interest as a means to identify therapeutic targets, particularly those involving cell proliferation, apoptosis and metabolism. The aberrant pathways currently under investigation include the following: Wnt/β-catenin signalling (activated in up to 50% of HCC) [70]; phosphatidylinositol-3-kinase/protein kinase B/mammalian target of rapamycin (PI3K/Akt/Mtor) (activated in 40–60% of HCC) [71]; Myc (activated in 30–60% of HCC) [72,73]; Hedgehog signalling (activated in 50–60% of HCC) [74,75]; and mesenchymal epithelial transition (MET) (activated in 30–40% of HCC) responsible for metastasis and migration [76]. These are excellent potential targets; however, the modulators of these pathways have yet to reach clinical practice.

## 6. Clinical Presentation of MASLD-HCC

Due to the issue of the late presentation of HCC, it is important to identify it early in its clinical course to enhance the potential for curative or ablative therapies. However, HCC surveillance is challenging, and the guidelines vary. For instance, the AASLD guidelines currently only recommend HCC surveillance in MASLD for patients with cirrhosis [2], the EASL guidelines recommend additional individual risk assessment in patients with F3 fibrosis [77] and the American Gastroenterological Association recommends surveillance in advanced fibrosis if at least two noninvasive testing modalities are concordant [78]. For patients with MASLD without advanced fibrosis or established cirrhosis, the individual risk of HCC development remains low (an incidence of 0.03 per 100 person years, 95% CI: 0.01–0.07 [79]). This coupled with the size of the population would make the cost of regular USS surveillance in this population prohibitive without further stratification. Furthermore, with the high rates of obesity in MASLD, there is an increased likelihood of central obesity, which potentially hampers the quality of USS imaging and reduces the detection of smaller lesions. There is also the potential for harm as a result of the overinvestigation of false-positive tests.

In cases of non-cirrhotic HCC, as individuals are generally not in a surveillance programme, lesions are more likely to be picked up incidentally or symptomatically (the latter in particular making curative treatment very unlikely) [58]. Whilst the liver function tends to be better preserved in MASLD-HCC patients without cirrhosis, the tumours tend to be more advanced at the time of presentation, resulting in poorer overall outcomes than other aetiologies of CLD [57,80]. This is due to patients having larger tumours at diagnosis and more likely to have a diffuse infiltrative pattern into the liver parenchyma [57]. They are also more likely to be older with more comorbidities, especially heart disease [32] and other components of metabolic syndrome. As a result of this, they are less likely to be offered curative surgical therapies, such as orthotopic liver transplantation (OLT) or resection [81], with fewer options for HCC-specific treatment [55], and, hence, they have worse outcomes.

## 7. Identification and Risk Stratification of MASLD Patients for HCC

The EASL policy statement published in April 2023 advocates for risk-based surveillance for hepatocellular carcinoma in cirrhosis [82]. A similar approach for patients in the pre-cirrhotic stage would allow for the screening of higher-risk patients in an economically viable fashion. However, there are currently no disease-specific, evidence-based strategies or reliable biomarkers to identify patients with non-cirrhotic MASLD at a higher risk of progression and HCC development. Effective, economically viable methods of identifying which patients with MASLD have significant fibrosis is challenging due to the scale of the patient group and the difficulty in predicting the rate of progression due to the complexity of the underlying disease.

There are a number of potential factors allowing MASLD patients to be categorised into high- and low-risk groups for HCC to allow for targeted surveillance. The combination of the presence of certain risk factors, as described above, could be used to identify patients who may benefit from earlier screening, similar to the PAGE-B and REACH-B scores in patients with chronic hepatitis B [83,84]. This form of precision medicine would require a well-validated scoring calculator with individualised risk thresholds to trigger screening for each patient. While no such current risk calculators exist for HCC screening, research is ongoing in this area, both in clinical studies and via machine learning [85,86]. For example, the GALAD score has been designed as a tool for earlier detection of HCC whichcombines gender, age and alpha-fetoprotein (AFP), des-carboxy-prothrombin (DCP) and AFP isoform L3 (AFP-L3) serum tests. This tool showed excellent diagnostic results in a retrospective case–control study compared to liver biopsy [87], and it has been validated independently with an optimal sensitivity of 91% and a specificity of 85% for HCC detection [88]. However, its usefulness as a surveillance biomarker still requires investigation.

The liquid biopsy is a diagnostic test performed on serum for multiple biomarkers related to tumour cells. The key markers are circulating tumour cells (CTCs) and circulating tumour DNA (ctDNA). CTCs are tumour cells shed from the original tumour and are identified via the presence of specific proteins (e.g., the presence of EpCAM and CK8/18/19 or the absence of CD45). The main limitation is the extremely low number of CTCs in blood samples, particularly in early cancer, and their very short half-lives. Circulating tumour DNA (ctDNA) are fragments of DNA released by tumour cells during apoptosis, and they contain the aberrant genetic information of the tumour, including the modification of the DNA, such as methylation signatures. However, alongside the detection of ctDNA is the dilution effect of cell-free DNA (cfDNA), which is similarly fragmented DNA and released by non-malignant apoptotic host cells. This limits the ability to isolate ctDNA (which can represent as little as 0.01% of the total cfDNA in the circulation) from non-malignant cfDNA unless searching for a specific mutation or methylation signature [89]. If detected, ctDNA can reveal genetic mutations or amplifications relevant to HCCs, such as P53, Wnt, β catenin and mutations relevant to the cellular response to oxidative stress (KEAP1, NFE2L2). These traces of tumour biomarkers have the potential to allow for a personalised medicine approach in which specific DNA changes determine the treatment protocol. Furthermore, as HCC is a disease that arises against the backdrop of distinct conditions, the biomarkers would need to be disease-specific. The identification of specific biomarkers in HCC arising in the setting of MASLD is an area of ongoing investigation.

## 8. Therapeutic Approaches to Reduce Progression to MASLD-HCC

There is significant interest in interventions that aim to potentially modulate the HCC risk profile of high-risk patients. The management of MASLD itself to prevent disease progression is likely to be key. A number of interventions have been shown to have the potential to reduce both the MASLD progression and HCC risk.

Lifestyle management strategies, such as dietary modifications aiming at weight loss and the treatment of the underlying metabolic syndrome, are the mainstays of MASLD therapy. However, lifestyle changes in particular can be challenging due to the need to maintain them from the medium to long term. Regular physical activity, and particularly aerobic and resistance training, have been shown to lead to a decrease in overall liver fat, independent of weight loss [18], and now form part of recommendations in recently published guidelines from the British Society of Gastroenterology/British Association for the Study of the Liver (BASL/BSG) on the management of MASLD 2023 [90]. In terms of the oncogenic risk, the pan-European EPIC cohort study (European Prospective Investigation into Cancer and Nutrition) demonstrated an association between physical activity (defined as 2 or more hours of vigorous activity per week) and a reduction in the HCC risk, with a hazard ratio of 0.50 (95% CI: 0.33–0.76) in the group that undertook vigorous physical activity [91]. Weight loss is the intervention most likely to be of benefit. With dietary changes alone, patients are advised to aim for a calorie deficit targeting 5–10% weight loss. While as little as 5% weight loss improves steatosis [17], 7–10% weight loss has been shown to improve histological endpoints such as the MASLD activity score and fibrosis, with a more than 10% body weight reduction associated with MASH resolution and an improvement in fibrosis by one stage [19]. However, the optimal dietary recommendation for MASLD is not known. Currently, the Mediterranean diet is the most widely recommended in view of its positive effects on cardiovascular risk [92] and has recently been shown to improve the intrahepatic lipid contents [93]. Weight management services should be considered if weight loss goals are not achieved with dietary changes alone. Bariatric surgery to achieve weight loss has also been investigated for its impact on the MASLD and HCC risk. A systematic review and meta-analysis of nine studies assessing the incidence of HCC post-bariatric surgery demonstrated a reduction in the HCC risk (an incidence rate ratio of 0.28; 95% CI: 0.18–0.42), although further analysis was limited by incomplete data on the prevalence of MASLD/MASH [94]. A more recent meta-analysis of 32 studies in which patients were followed up for at least 3 years post-bariatric surgery versus lifestyle changes also only identified a reduced overall risk of all cancers, with a similar reduction in the future incidence of HCC (relative risk: 0.35; 95% CI: 0.22–0.55), although again there was limited analysis of the liver disease status. A more focused propensity-matched analysis of patients post-bariatric surgery and obese controls from a single centre (a total of 4112 patients) demonstrated reduced new-onset MASH (6% vs. 10%, *p* < 0.0001) and HCC (0.05% vs. 0.34%, *p* = 0.03) over a median 7.1-year follow-up period.

A detailed review of the pharmacological therapies for MASLD and their supporting evidence is beyond the scope of this work. Broadly, the aim is for the robust management of the cardio-metabolic risk factors, including dyslipidaemia, hypertension and type 2 diabetes. There are currently no approved pharmacological therapies specifically for the treatment of MASLD. However, of particular interest are agents from two anti-diabetic medication classes: PPARγ agonists and GLP-1 receptor agonists. Pioglitazone is a PPARγ agonist that acts as an insulin sensitiser and has been associated with a significant reduction in insulin resistance, steatosis and hepatic histological improvements (including inflammation and ballooning). It has been shown to reduce fibrosis progression and the development of HCC in vitro and in murine models [95,96], with promising results in human case–control and cohort studies to date [97,98,99]. However, the significant side-effect burden of this class of drug, including bladder cancer risk, cardiovascular events and weight loss, warrants further review. The GLP-1 receptor agonist semaglutide is currently part of Phase 3 trials as a therapy for MASH, following promising results from earlier studies regarding improvements in MASH without worsening fibrosis [22]. A recently published cohort study was the first to compare GLP-1 receptor agonists vs. long-acting insulin therapy in patients with type 2 diabetes mellitus, demonstrating a lower risk of cirrhosis and HCC in the GLP1 receptor agonist group [100].

Within the armoury of anti-diabetic medication, whilst tight glycaemic control has been shown to improve MASLD, overall, metformin appears to have a more significant chemoprotective effect against HCC than other classes of drugs, including insulin. This is thought to be related to its role in downregulating the MTOR pathway. It has been shown to inhibit cancerous cell growth via the induction of cell cycle arrest and the enhancement of apoptosis [101]. A meta-analysis of five studies of patients with diabetes has shown a lower risk of developing HCC in patients treated with metformin (odds ratio: 0.38; 95% CI: 0.24–0.59), albeit with significant heterogeneity, and without significant detail regarding the presence or absence of MASLD [102]. A further systematic review comparing metformin vs. insulin/sulphonylureas also demonstrated a reduced risk of HCC with metformin, with an OR of 0.47 (CI: 0.28–0.80), which was not seen with the other therapies [103]. Again, there were no liver-specific data. One further case–control study showed a dose-dependent effect in the protective effects of metformin against HCC in diabetic patients, with a 7% reduction in the HCC risk per incremental year of metformin therapy [104]. In this study, 3.5% of the HCC patients and 0.3% of the controls had liver cirrhosis not attributed to alcohol or viral hepatitis. However, there is the potential for bias in the prescription of this drug, which is traditionally avoided in patients with more advanced liver disease.

Statins are also of great interest for their role in modulating the HCC risk due to their antioxidative, anti-inflammatory, endothelial function and anti-fibrotic properties. Several meta-analyses have demonstrated a reduction in the incidence of HCC in patients taking statins compared to controls, although, again, there were insufficient data for MASLD-specific analysis [105,106,107]. A further meta-analysis of 24 studies of patients with HCC identified a similar protective role of statin exposure, which appeared to remain significant upon a subgroup analysis of patients with cirrhosis [108].

Overall, there is a need for research specifically focused on patients with MASLD undergoing such interventions to assess their efficacy, as most studies to date have not delineated this particular patient group in detail.

## 9. Therapeutics in HCC

Treatment options for HCC include surgical, locoregional and systemic therapies. The choice of therapy is guided by the tumour burden, liver function and WHO performance status. The traditional tumour–node–metastasis classification for the staging of solid-organ malignancy is less useful in HCC, partly due to the need to incorporate the degree of liver dysfunction into the decision-making process. While there are multiple proposed alternative staging systems, the Barcelona Liver Clinic Cancer (BCLC) (shown in Figure 2) is the most commonly used and is recommended by the AASLD [109] and EASL [77] to help guide treatment.

### 9.1. Curative Treatments

Surgical resection offers a curative treatment for non-cirrhotic patients with single localised lesions. Patients with cirrhosis are considered candidates for resection if they have a solitary nodule without evidence of extra-hepatic spread and are well compensated (Child Pugh A), with no evidence of clinically significant portal hypertension. Individuals with MASLD may have comorbidities affecting their risk for surgery, which represents an additional barrier to curative treatment for these individuals. Orthoptic liver transplantation (OLT) is a curative option for patients not eligible for resection (e.g., due to evidence of liver dysfunction or multinodular disease) who meet the Milan criteria (one lesion < 5 cm or from two to three lesions between 3 and 5 cm). The risk of recurrence of HCC post-OLT is significantly lower than in the postresection group, at 10% at 5 years compared to 50–70% [111], although the prognosis of recurrent HCC in the post-OLT group is poorer. However, a shortage of organ donors remains a significant limitation. While being assessed for OLT or on the waiting list, patients should be offered bridging therapy in the form of neoadjuvant locoregional therapy (e.g., transarterial chemoembolisation (TACE) or ablation). Ablation is a further treatment with curative intent for single lesions in patients with preserved liver function who decline or are not eligible for the surgical treatments described above. This can be performed via radiofrequency (RFA) or thermal ablation.

### 9.2. Non-Curative Treatments

In patients with BCLC stage B HCC, transarterial chemoembolisation (TACE) is recommended. While not curative, TACE offers a significant improvement in overall survival (OS) compared to the best supportive care. Combining TACE with systemic therapy has not so far shown significant improvement compared to TACE alone but further clinical trials are ongoing. Additional locoregional treatments include the stereotactic body radiotherapy “cyberknife” and targeted chemotherapy using Yttrium microspheres [112,113].

Systemic treatments are currently offered to patients with BCLC stage C disease, BCLC stage B disease unsuitable for the treatments described above or progressive disease following locoregional therapy. Another potential role for systemic therapy is as adjuvant treatment, as the postresection risk of recurrence remains high at 50–70% at 5 years postoperatively [114,115]. The currently approved systemic therapies can be classified into either anti-angiogenic targeted therapies (mTKIs and monoclonal anti-angiogenic antibodies) and immune checkpoint inhibitors. The AASLD recommends adjuvant immune checkpoint inhibitor-based systemic therapy in patients deemed at high risk for recurrence after resection or ablation.

Multi-tyrosine kinase inhibitors (mTKIs)

These inhibit multiple protein kinases, including the VEGF receptors. Sorafenib was the first systemic therapy to offer a survival benefit for patients not eligible for surgical or locoregional therapy. The SHARP trial compared sorafenib to a placebo, with a median OS of 10.9 vs. 7.9 months but a significant side-effect burden [116]. Lenvatinib, which has activity against multiple kinases, including VEGF receptors, has since been shown to be non-inferior to sorafenib in the 2018 REFLECT study and is now considered a first-line alternative to sorafenib [117]. Cabozantinib and regorafenib are also being assessed for efficacy as second-line treatments.

Immune checkpoint inhibitors

Immune checkpoint inhibitors (ICIs) target specific receptors within the inflammatory pathways of the tumour microenvironment with the aim of restoring anti-tumour T-cell activity. The principle behind these treatments is that there are reactive tumour-specific T cells that can be re-awakened to generate an anti-cancer immune response. The ICIs currently available include programmed death-1 (PD-1) inhibitors, programmed death ligand-1 (PD-L1) inhibitors and cytotoxic T lymphocyte antigen (CTLA-4) inhibitors.

PD-1 receptors are expressed on a variety of immune effector cells, including activated T cells, NK cells and dendritic cells. Tumour cells have been shown to overexpress PD-L1, leading to PD-1 activation in tumour-infiltrating lymphocytes, thereby dysregulating immune surveillance. HCCs with higher expressions of PD-L1 are associated with a poorer prognosis and more aggressive tumours [118,119,120]. The PD-1 inhibitors currently licensed for use in HCC are nivolumab and pembrolizumab. CTLA-4 is a transmembrane receptor expressed on the surfaces of activated T cells that can induce the unresponsiveness of T cells via the out-competition of part of the co-stimulatory signal required for full T-cell activation [121]. The CTLA-4 inhibitors of current clinical interest are ipilimumab and tremelimumab.

Multiple clinical trials assessing these immune checkpoint inhibitors as single agents or combination therapy are ongoing. In a Phase 2 trial, the KEYNOTE-224 study demonstrated that the anti-PD-1 antibody pembrolizumab was safe and well tolerated in patients previously treated with sorafenib. Higher combined positive score (CPS) (calculated as the percentage of the total viable tumour cells that express PD-L1) was associated with a higher ORR and longer PFS [122]. This suggests that immunologically active tumours are more responsive to immunotherapy.

The IMbrave150 trial was a landmark trial in HCC. In this study, the combination of the PD-L1 inhibitor atezolizumab and the monoclonal anti-angiogenic antibody bevacizumab was compared to sorafenib and demonstrated a longer median OS (19.2 months vs. 13.4 months) [123,124]. Due to the risk of variceal bleeding noted during the study, band ligation is recommended prior to initiation. This combination is now licensed by both the FDA and NICE as an alternative first-line therapy to sorafenib and lenvatinib. However, the side effects from this treatment remain problematic [125]. More recently, the CARES-310 study of camrelizumab (anti-PD-1) and revoceranib (a highly selective VEGFR2 tyrosine kinase inhibitor) also demonstrated superior progression-free and OS in comparison to sorafenib. In this study, the median survival was 22.1 months with the camrelizumab/revoceranib combination in comparison to 15.2 months with sorafenib. Serious adverse events were noted in 24% of patients in the camrelizumab/revoceranib group [126]. The HIMALAYA study Phase III trial compared the PD-L1 inhibitor durvalumab in combination with the CTLA4 inhibitor tremelimumab (STRIIDE) versus sorafenib, with an improved OS (16.4 months vs. 13.7 months) [127]. This combination was approved by the FDA in 2022. Interestingly, durvalumab monotherapy was non-inferior to sorafenib. The area of ICI combination is paving the way for further HCC therapeutics; however, due to their mechanisms of action in boosting immune reactivities, they are not recommended in autoimmune liver disease or in the post-transplant population.

### 9.3. Challenges in MASLD-HCC Treatment

While the EASL and AASLD guidelines do not recommend the selection of a treatment pathway based on the aetiology of underlying liver disease, these guidelines have historically been based on patients with viral hepatitis. Furthermore, as described earlier, patients with MASLD-HCC are more likely to have other significant comorbidities related to metabolic syndrome, which can complicate decisions regarding the choice of treatment and potentially worsen the peri-operative risk profile.

Overall, patients with MASLD-HCC are likely to be older patients with poorer performance statuses (PSs) [32]. They are likely to have larger tumours at diagnosis but are less likely to have established cirrhosis or clinically significant portal hypertension [30,106], and therefore they are less frequently enrolled in a surveillance programme. This results in most patients being diagnosed at more advanced stages of disease (BCLC C or D), with a lower opportunity for curative treatment [128]. Of note, the BCLC staging does not account for non-cirrhotic HCC, and therefore patients may be staged as more “advanced” disease in MASLD-HCC compared to other aetiologies purely on the basis of the tumour size/number.

In terms of outcomes, in studies focusing on curative therapies in HCC (OLT, resection, RFA), MASLD patients were older with lower MELD scores [129]. There was no difference in the recurrence-free survival at a median of 50 months of follow-up. Reassuringly, these patients had improved OS independent of other clinical features compared to those with HCV and ARLD. Similar findings were noted in a meta-analysis focusing again on patients offered curative treatment [130]. However, a further study in Germany that looked at a cohort of over 1000 MASLD-HCC patients, irrespective of the treatment offered, noted worse OS in MASLD compared to other aetiologies [80]. This is most likely due to the selection of patients with less advanced disease in the curative studies. Immediate post-OLT complications in terms of peri-operative events, such as higher rates of infections and longer lengths of stay, are observed in patients transplanted for MASLD compared to other aetiologies, particularly those with multiple components of metabolic syndrome [131]. Reassuringly, the mortality and graft survival rates appear to be similar between aetiologies. The recurrence of MASH post-OLT is also increasingly common [132]. Table 4 summarises several studies comparing the outcomes of patients with MASLD-HCC with those of patients with other aetiologies of liver disease across the spectrum of surgical and interventional therapies.

In terms of systemic therapies, there is increasing evidence that the disease aetiology influences the treatment response [137]. However, data specific to MASLD-HCC are limited. The pivotal SHARP study included only patients with viral hepatitis and alcohol-related liver disease. A recent international cohort study of patients treated with sorafenib did not show any difference in the outcomes (OS, sorafenib-specific survival and rates of toxicity) between MASLD-HCC and other aetiologies but had only a small (3.6%) MASLD-HCC subgroup [138].

There are suggestions based on pre-clinical models that MASLD-HCC does not respond as effectively to immune checkpoint inhibitor therapy, but there is a need for further confirmatory studies on this subgroup. In individuals with chronic viral hepatitis, the immune system is already dysfunctional and this may predispose them to altered treatment responses in vivo. A meta-analysis of IMbrave150, KEYNOTE240 and Checkmate459 showed that whilst immunotherapy improved survival in the overall cohort, when divided into viral-hepatitis-related HCC vs. non-viral-HCC, the survival benefit was not maintained in the non-viral group [64]. However, none of these key studies distinguished MASLD-HCC as a separate aetiology of liver disease at baseline, with the cohorts being divided according to the presence or absence of viral hepatitis. A small study looking at anti-PD(L)1 immunotherapy compared the outcomes of MASLD-HCC patients (n = 13) to those with other aetiologies (n = 117) and demonstrated reduced OS in the MASLD-HCC group [64]. Interestingly, a post hoc analysis of IMbrave150 reviewed 279 of the 336 patients in the immunotherapy arm of the original study to stratify them into MASLD viral hepatitis and ARLD-related HCC and did not show statistically significant differences in the overall response rate, progression-free survival (PFS) or OS between aetiologies [139]. The HIMALYA (~42% non-viral participants) and CARES-310 (~16% non-viral participants) trials did not selectively report outcomes in MASLD-HCC, and therefore the topic of the ICI response in MASLD-HCC requires further investigation.

A retrospective cohort study looking at the therapeutic efficacy of lenvatinib has shown similar OS and PFS between subgroups of MASLD-HCC vs. all other aetiologies [140]. A larger meta-analysis of eight studies assessing the disease response to all systemic therapies concluded that ICI therapy offers more benefit in viral than non-viral HCC, while the efficacies of TKI and anti-VEGF agents appear to be independent of the aetiology of underlying liver disease [141]. None of these studies made further distinctions within the “non viral HCC” group in terms of MASLD, alcohol or other aetiologies of chronic liver disease. Overall, patient stratification for HCC treatment is still rudimentary, without molecular considerations, and more rational therapeutic strategies are currently being tested in clinical trials using existing therapeutics.

## 10. Conclusions and Future Directions

MASLD-HCC is increasing in parallel with the MASLD epidemic and is proving to be a complex disease that is often diagnosed at an advanced stage due to a significant proportion manifesting before the standard HCC surveillance thresholds are reached. Further work is needed to generate well-validated scoring systems to identify high-risk patients for the initiation of HCC surveillance, as well as the refinement of diagnostic methods to detect HCC at an earlier stage. Treatment for MASLD-HCC, as for all HCCs, is challenging with the MASLD population, who have specific difficulties related to their comorbidities. A bespoke approach to MASLD-HCC still needs to be identified in order to better manage this group of patients.

## Figures and Tables

**Figure 1 cancers-16-00259-f001:**
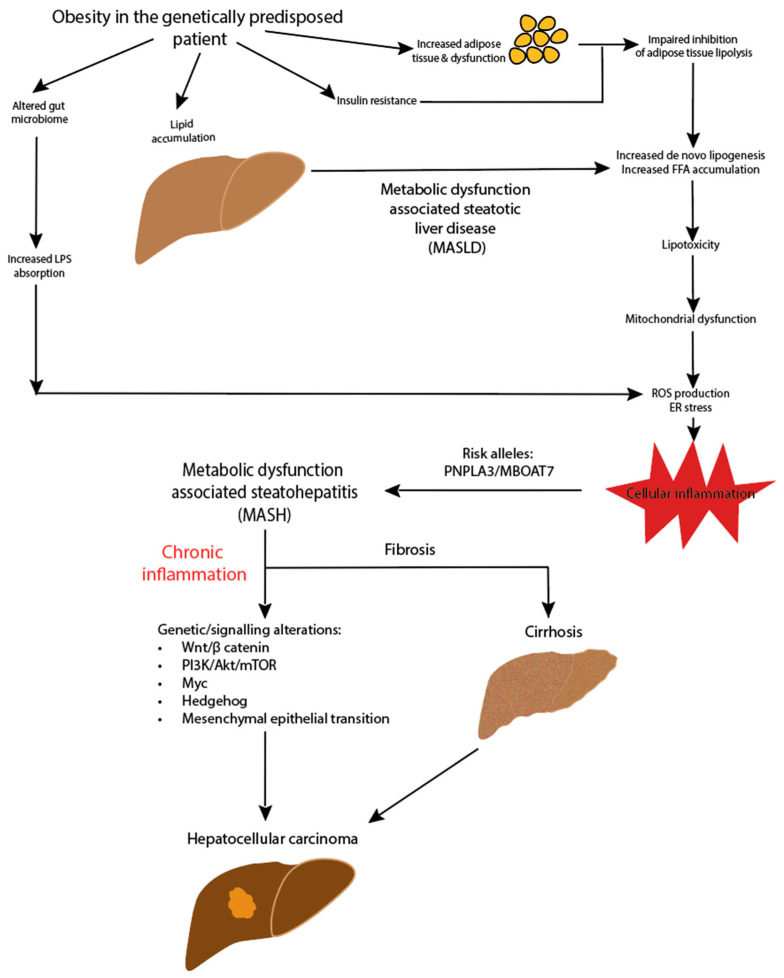
Pathogenesis of MASLD-HCC.

**Figure 2 cancers-16-00259-f002:**
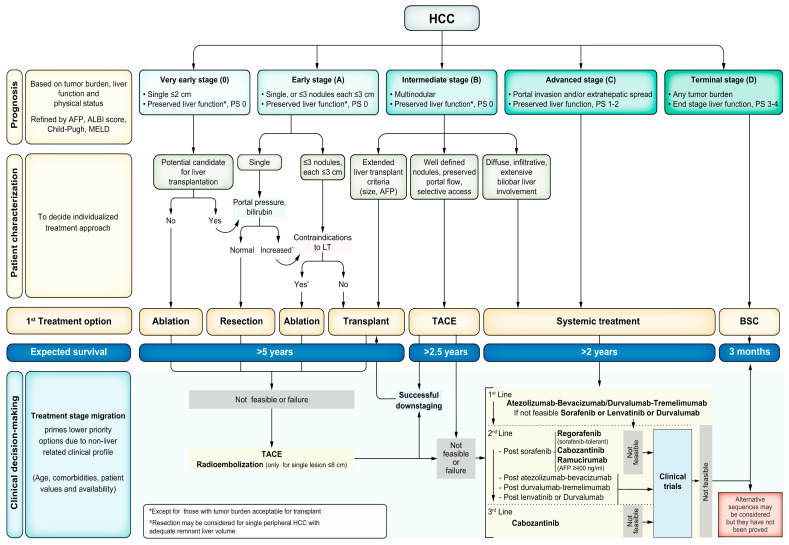
Updated Barcelona Clinic Liver Cancer Staging System 2022. Abbreviations: AFP, alpha fetoprotein; ALBI, albumin–bilirubin; BSC, best supportive care; ECOG-PS, Eastern Cooperative Oncology Group performance status; HCC, hepatocellular carcinoma; LT, liver transplant; MELD, Model for End-Stage Liver Disease; TACE, transarterial chemoembolisation. Reprinted with permission from Reig et al. [110].

**Table 2 cancers-16-00259-t002:** Studies of MASLD cohorts comparing the incidence of HCC with and without cirrhosis.

Ref.	Study Design	Years Studied	Country	Size of Cohort	HCC Rates	Risk Factors	Duration of Follow-Up
[39]	Retrospective cohort study of MASLD and control patients from 130 facilities in the Veterans Health Administration	2004–2015	U.S.	296,707 MASLD patients with 296,707 matched controls	MASLD: 0.21/1000 PYs Control: 0.02/1000 PYs		9 years (SD: 2.2)
				Cirrhosis (80%)	10.6/1000 PYs	Age > 45 years Male Hispanic ethnicity	
				Without cirrhosis	0.08/1000 PYs		
[51]	Retrospective cohort study of patients with cirrhosis diagnosed between 2001 and 2014 in the Veterans Affairs healthcare system	2001–2017	U.S.	116,404 patients	2/100 PYs	Age Male Hispanic ethnicity AFP ALP AST/ALT ratio Serum albumin Platelet count	4.3 years
				MASLD (15%)	9/100 PYs	T2DM	
				HCV (45%)	3.3/100 PYs		
				ALD (31%)	0.86/100 PYs	T2DM Elevated BMI	
[52]	Retrospective cohort study of patients with MASLD steatosis only	Pre 2012	Japan	6508 patients	0.43/1000 PYs	Age T2DM Platelet count AST	5.6 years
[53]	Retrospective cohort study of patients with cirrhosis	2003–2007	U.S.	510 patients		Age Alcohol intake	3.2 years (1.7–5.7)
				MASH: 195	Yearly cumulative incidence: 2.6% p.a.		
				HCV: 315	4.0% p.a.		
[54]	Retrospective cohort study of patients undergoing health check ups between 2004 and 2005 at a tertiary referral hospital	2004–2015	Korea	25,947 patients MASLD: 33.6% (NB: patients with cirrhosis excluded)	782.9/100,000 PYs	High MASLD fibrosis score (NFS) and high fibrosis-4 (FIB-4) score	7.5 years (3.2–9.3)

Abbreviations: AFP, alpha fetoprotein; ALP, alkaline phosphatase;; ALT, Alanine transaminase; AST, aspartate aminotransferase; HCC, hepatocellular carcinoma; HCV, hepatitis C virus; MASH, metabolic dysfunction-associated steatohepatitis; MASLD, metabolic dysfunction associated steatotic liver disease; PYs, person years; T2DM, type 2 diabetes mellitus.

**Table 3 cancers-16-00259-t003:** Studies of HCC cohorts: proportions of MASLD-HCC patients without cirrhosis.

Ref.	Study Design	Years Studied	Country	Size of Cohort	Proportion with Condition	Proportion without Cirrhosis
[55]	Retrospective observational study of patients with confirmed HCC in the U.S. Veterans Administration	2005–2010	U.S.	1500 patients		Without cirrhosis overall: 13%
					MASLD: 8%	34.6%
[56]	Retrospective observational study of patients with histologically proven MASH who developed HCC	1993–2010	Japan	87 patients	MASH: 100%	49%
[57]	Multi-centre prospective observational study of patients with HCC with either HCV or MASLD	2010–2012	Italy	756 patients	HCV: 81%	2.8%
					MASLD: 19%	46.2%
[30]	Retrospective evaluation of patients with verified HCC via U.S. healthcare insurance database	2002–2008	U.S.	4406 patients		Without cirrhosis overall: 25%
					MASLD: 59%	54%
[32]	Retrospective population-based study of patients within Medicare-linked HCC registry	2004–2009	U.S.	4929 patients	MASLD: 9% annual increase from 2004 to 2009 (from 14.4 to 20.3%)	Not able to assess due to data
[58]	Prospective cohort study of patients with HCC referred to a single tertiary liver unit	2000–2010	U.K.	632 patients	MASLD: 35% in 2010, increased from none/undefined in 2000	22.5%

Abbreviations: HCC, hepatocellular carcinoma; HCV, hepatitis C virus; MASH, metabolic dysfunction-associated steatohepatitis;.MASLD, metabolic dysfunction associated steatotic liver disease.

**Table 4 cancers-16-00259-t004:** Outcomes of MASLD-HCC from surgical and interventional therapies.

Treatment	Study Details	Proportion with MASLD	Outcomes	Reference
Resection	Meta-analysis of 15 cohort studies/7226 patients	MASLD: 19.5%	MASLD: better DFS and OS	[133]
Resection	Meta-analysis of nine studies/5579 patients	MASLD: 20.1%	MASLD: better DFS and OS	[130]
	Retrospective cohort study via Medicare database: 17,664 patients undergoing curative HCC treatment	MASLD: 33.4%	MASLD improved survival postresection	[128]
OLT	European Liver Transplant Registry analysis 2002–2016/68,950 transplant recipients of all aetiologies (20,195 patients with HCC)	MASLD: 4% MASLD HCC: 5.6% of HCC cohort	MASLD-HCC vs. ARLD: 10-year post-OLT survival of 46.9% vs. 51.8% slightly worse but no difference compared to HCV (48.2%)	[5]
	Retrospective cohort design study in two centres in Toronto and San Franciso: OLT for HCCs of all aetiologies in 2004–2014: 929	MASH: 6.5%	No difference between MASH and non-MASH patients at 1-, 3- and 5-year survival	[134]
	Retrospective cohort study via Medicare database: 17,664 patients undergoing curative HCC treatment	MASLD: 33.4%	MASLD: worse median survival than non-MASLD	[128]
RFA	Multi-centre retrospective cohort study of 520 HCC patients of all aetiologies	MASLD: 12.6%	MASLD: no difference in OS, tumour recurrence	[135]
	Retrospective cohort study via Medicare database: 17,664 patients undergoing curative HCC treatment	MASLD: 33.4%	No significant difference	[128]
All curative therapies	Meta-analysis of nine studies/5579 patients	MASLD: 20.1%	MASLD: better OS and DFS (DFS not statistically significant)	[130]
TACE	Single-centre retrospective cohort study of 220 patients treated for 353 HCCs of all aetiologies between 2011 and 2016, U.S.	MASLD: 13.6%	MASLD: no difference in OS, time to progression or complication rates	[136]

Abbreviations: ARLD, alcohol-related liver disease; DFS, disease free survival; HCC, hepatocellular carcinoma; HCV, hepatitis C virus; MASH, metabolic dysfunction-associated steatohepatitis; MASLD, metabolic dysfunction associated steatotic liver disease; OLT, orthoptic liver transplant; OS, overall survival; RFA, radiofrequency ablation; TACE, transarterial chemoembolization.

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
