# Peer review of "MASLD and the Development of HCC: Pathogenesis and Therapeutic Challenges"

_cancers, 2024, doi:10.3390/cancers16020259_

Round 1
Reviewer 1 Report
Comments and Suggestions for Authors
The review is undoubtedly well written although there are several other reviews on the same topic so i have some concerns on the novelty of the paper.
I liked the long section on therapeutic outcomes in NAFLD-related HCC but i would like to see also another section on the treatments to prevent the occurrence of HCC in NAFLD patients. In this regard cite the recent SRMA: PMID: 33721336 )
Some figures would improve the quality of the manuscript
Consider the use of the term MAFLD instead of NAFLD in the manuscript
Reviewer 2 Report
Comments and Suggestions for Authors
The authors have reviewed NAFLD and HCC development: pathogenesis and treatment challenges.
As the title suggests, the literature reviews are well-organized and helpful to readers.
Minor points.
The international liver societies like AASLD and EASL have changed the terms regarding fatty liver disease nomenclature. Metabolic dysfunction associated steatotic liver disease (MASLD) is replaced term for NAFLD. Metabolic dysfunction-associated steatohepatitis (MASH) is also replaced term for NASH. Therefore, the terms should be changed according to it.
It is considered that there are no additional issues related to the manuscript.
Comments on the Quality of English Language
None.
Round 2
Reviewer 1 Report
Comments and Suggestions for Authors
The revised version of the paper is OK. Thank you!